# Human visual gamma for color stimuli

**Benjamin J Stauch[1,2,3]\*, Alina Peter[1,2], Isabelle Ehrlich[1,4], Zora Nolte[1], Pascal Fries[1,2,5]\***

[1]Ernst Strüngmann Institute (ESI) for Neuroscience in Cooperation with Max Planck Society, Frankfurt, Germany; [2]International Max Planck Research School for Neural Circuits, Frankfurt, Germany; [3]Brain Imaging Center, Goethe University Frankfurt, Frankfurt, Germany; [4]Department of Psychology, Goethe University Frankfurt, Frankfurt, Germany; [5]Donders Institute for Brain, Cognition and Behaviour, Radboud University Nijmegen, Nijmegen, Netherlands

**Abstract** Strong gamma-band oscillations in primate early visual cortex can be induced by homogeneous color surfaces (Peter et al., 2019; Shirhatti and Ray, 2018). Compared to other hues, particularly strong gamma oscillations have been reported for red stimuli. However, precortical color processing and the resultant strength of input to V1 have often not been fully controlled for. Therefore, stronger responses to red might be due to differences in V1 input strength. We presented stimuli that had equal luminance and cone contrast levels in a color coordinate system based on responses of the lateral geniculate nucleus, the main input source for area V1. With these stimuli, we recorded magnetoencephalography in 30 human participants. We found gamma oscillations in early visual cortex which, contrary to previous reports, did not differ between red and green stimuli of equal L-M cone contrast. Notably, blue stimuli with contrast exclusively on the S-cone axis induced very weak gamma responses, as well as smaller event-related fields and poorer change-detection performance. The strength of human color gamma responses for stimuli on the L-M axis could be well explained by L-M cone contrast and did not show a clear red bias when L-M cone contrast was properly equalized.

**\*For correspondence:**
benjamin.stauch@esi-frankfurt.de (BJS);
pascal.fries@esi-frankfurt.de (PF)

## Editor's evaluation

Previous work has shown that presentation of uniform fields of color can induce strong gamma rhythms in visual cortex of monkeys and humans, particularly red fields. However, prior work has rarely considered the effectiveness of the colored visual stimulus at driving the input to visual cortex. This study uses MEG measurements from human visual cortex and shows that this preference for red is reduced when colored stimuli are likely to drive similar levels of input to visual cortex.

## Introduction

In early visual cortex of human and non-human primates, visual stimuli often induce rhythmic activity in the gamma band (*Hoogenboom et al., 2006*; *Jia et al., 2011*; *Kreiter and Singer, 1992*). Gamma oscillations and their coherence between neuronal populations have been related to attentional selection (*Bauer et al., 2006*; *Bosman et al., 2012*; *Fries et al., 2001*; *Magazzini and Singh, 2018*), directed information flow (*Bastos et al., 2015*; *Besserve et al., 2015*; *Michalareas et al., 2016*; *van Kerkoerle et al., 2014*), stimulus learning (*Brunet et al., 2014*; *Peter et al., 2021*; *Stauch et al., 2021*), and further cognitive functions (for reviews, see *Fries, 2015*; *Singer, 2018*).

In recent years, research on visually induced gamma has increasingly focused on the fundamental question of which visual stimulus features determine gamma amplitude. Especially strong gamma oscillations have been observed for high-contrast black and white stimuli that extend beyond the size

of single-neuron receptive fields (*Gieselmann and Thiele, 2008*; *Jia et al., 2011*) and whose features are contiguous over space (*Brunet and Fries, 2019*; *Hermes et al., 2019*; *Lowet et al., 2017*; *Uran et al., 2022*; *Vinck and Bosman, 2016*). Likely, this dependence exists because extensive and uniform stimuli synchronize larger populations of neurons into a common gamma rhythm, thereby generating a clearer oscillatory population signal (*Gray et al., 1989*; *Lowet et al., 2015*; *Lowet et al., 2017*).

Large homogeneous patches of color have also been shown to induce strong gamma oscillations, both in the early visual cortex of macaques (*Brunet et al., 2015*; *Peter et al., 2019*; *Rols et al., 2001*; *Shirhatti and Ray, 2018*) and humans (*Bartoli et al., 2019*; *Li et al., 2022*; *Perry et al., 2020*). Interestingly, red stimuli have been reported to induce especially strong gamma-band responses in several of these reports (*Bartoli et al., 2019*; *Li et al., 2022*; *Perry et al., 2020*; *Rols et al., 2001*; *Shirhatti and Ray, 2018*). It has been speculated that this might be due to a higher ecological relevance of the color red in natural environments (*Bartoli et al., 2020*), and it was recently related to potentially faster adaptation of the M-cone pathway (*Peter et al., 2019*). However, when comparing across different stimuli, the strength of input to the neuronal populations generating the investigated gamma signal should be controlled, because gamma-power scales with the strength of excitatory synaptic input (*Hadjipapas et al., 2015*; *Henrie and Shapley, 2005*; *Lewis et al., 2021*). Color-induced gamma has mostly been studied in area V1 (see above). In primates, feedforward driving input to V1 originates mostly from the LGN (*Felleman and Van Essen, 1991*; *Sherman, 2005*), including for color stimuli (*Chatterjee and Callaway, 2003*; *Shapley, 2019*). Note that in awake primates, the LGN itself does not show visually induced gamma (*Bastos et al., 2014*; tested with grating stimuli), such that V1 is likely the first stage in the primate visual system generating gamma.

Earlier work has established in the non-human primate that LGN responses to color stimuli can be well explained by measuring retinal cone absorption spectra and constructing the following cone contrast axes: L+M (capturing luminance), L-M (capturing redness vs. greenness), and S-(L+M) (capturing S-cone activation, which corresponds to violet vs. yellow hues). These axes span a color space referred to as DKL space (*Derrington et al., 1984*). This insight can be translated to humans (for recent examples, see *Olkkonen et al., 2008*; *Witzel and Gegenfurtner, 2018*), if one assumes that human LGN responses have a similar dependence on human cone responses. Recordings of human LGN single units to colored stimuli are not available (to our knowledge). Yet, sensitivity spectra of human retinal cones have been determined by a number of approaches, including ex vivo retinal unit recordings (*Schnapf et al., 1987*), and psychophysical color matching (*Stockman and Sharpe, 2000*). These human cone sensitivity spectra, together with the mentioned assumption, allow to determine a DKL space for human observers. To show color stimuli in coordinates that model LGN activation (and thereby V1 input), monitor light emission spectra for colored stimuli can be measured to define the strength of S-, M-, and L-cone excitation they induce. Then, stimuli and stimulus background can be picked from an equiluminance plane in DKL space.

Because spectral peaks of cone opsin sensitivity show variability over individuals (*Neitz and Jacobs, 1986*; *Neitz and Neitz, 2011*), recruiting a larger sample than is possible in human patient or macaque studies is necessary to measure the general, population-wide relationship between stimulus color and induced gamma oscillations in the early visual cortex.

In this study, we recorded MEG in 30 human participants while presenting them with uniform, circular color disks picked from a DKL equiluminance plane on an equiluminant background. Crucially, their cone contrasts on the L-M and the S-(L+M) axes were equalized. We analyzed the MEG recordings at the source level (*Gross et al., 2001*; *Van Veen et al., 1997*) and found that color-induced gamma-band responses localized to early visual cortex with a peak in area V1, confirming that color gamma responses can be studied non-invasively in humans. With colors that had identical luminance and identical color contrast in DKL space, we could not confirm stronger gamma responses for red stimuli. Instead, equiluminant colors with comparable absolute cone contrasts on the L-M axis induced gamma-power responses of equal strength in area V1, independent of their redness. Colors on the S-(L+M) axis induced weaker gamma power, especially for colors strongly activating only the S-cone.

## Results

Trial structure, stimuli, and stimulus color coordinates are shown in *Figure 1* and described in detail in Methods. In brief, participants fixated a central fixation spot on a gray background. After a 1.2 s

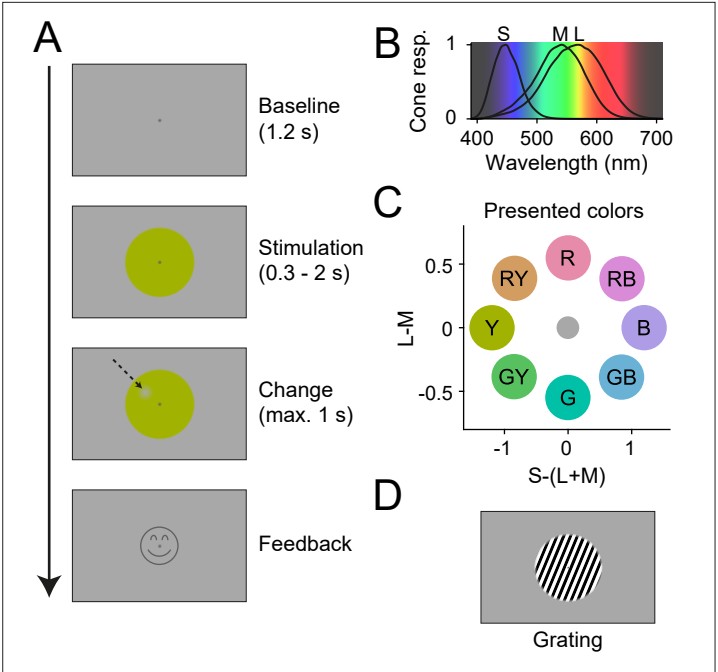

**Figure 1.** Experiment design. (**A**) Temporal structure of one trial. After fixation onset, a gray baseline was shown for 1.2 s, followed by 0.3–2 s of stimulation with a stimulus disk. Then, a change spot appeared at a random position on the disk (indicated here with an arrow, not visible in the actual experiment), which the participants needed to report. Upon correct report, a smiley was shown. (**B**) Human cone sensitivity spectra (*Stockman and Sharpe, 2000*) plotted on an estimate of perceptual wavelength color. (**C**) DKL coordinates for all eight stimulus colors shown in this study, relative to the background. Letters indicate stimulus labels used in Results. Note that color hues of these color disks, when displayed on a noncalibrated monitor or printout, will not fully correspond to the hues shown in the study. (**D**) To compare the strength of induced gamma responses between color and grating stimuli, some trials showed a grating instead of a color stimulus.

The online version of this article includes the following source data for figure 1:

**Source data 1.** Stimulus color coordinates.

baseline, they were presented with a stimulus disk that was centered on the fixation spot and had a diameter of 10 deg visual angle (dva) (*Figure 1A*) or with a luminance grating (*Figure 1D*). After 0.3–2 s, a change occurred at a random position on the disk, with the shape of a Gaussian blob (3.7 dva diameter). For colored disks, the change was a small decrement in color contrast, for gratings a small decrement in luminance contrast. In both cases, the decrement was continuously QUEST staircased (*Watson and Pelli, 1983*) per participant and color/grating to 85% correct detection performance. Subjects then reported the side of the contrast decrement relative to the fixation spot as fast as possible (max. 1 s), using a button press. Feedback was given as a smiley on correct trials, then the next trial commenced upon fixation.

The eight colors were sampled from an ellipse on a DKL equiluminance plane, such that they were DKL equiluminant to each other and to the background (*Figure 1C*). Color coordinates are given in *Figure 1—source data 1*. To simplify reporting, we will refer to colors on the L-M axis as 'red' and 'green', to colors on the S-(L+M) axis as 'blue' and 'yellow', and to colors with components from both axes as 'red-blue', 'red-yellow', 'green-blue', and 'green-yellow'. Note that these labels do not fully correspond to subjective perceptual labels: For example, monochromatic light strongly activating S-cones but not L-/M-cones would look purple to the observer, as can be seen from the spectral sensitivities of the S/M/L-cones (*Figure 1B*, *Stockman and Sharpe, 2000*).

Note that the colors were constrained to equalized cone contrast levels, while the grating was chosen to generate a maximally strong gamma signal for comparison. Differences in the strength of neuronal responses between grating and colors are therefore difficult to interpret, because it is unclear how input strength could be equalized.

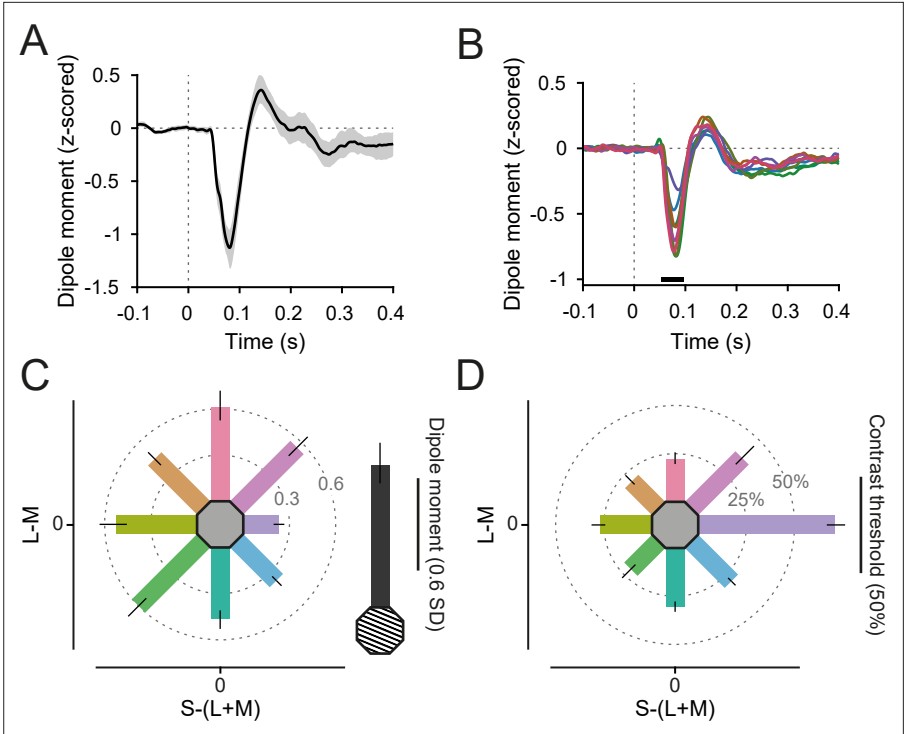

**Figure 2.** Event-related field (ERF) responses. (**A**) Grating-evoked ERF, averaged over V1 dipoles and participants. Error region shows 95% CI over participants. (**B**) Color-evoked ERFs, averaged over V1 dipoles and participants. Black bar indicates significant differences across colors, $t_{max}$ corrected for multiple comparisons. Line colors reflect stimulus colors, but have been darkened for readability. (**C**) ERF N70 dipole moment (relative to baseline variability), averaged over V1 dipoles and participants. Error bars represent 95% CIs over participants, bar orientation represents stimulus orientation in DKL space. In gray, the average grating-induced ERF dipole moment is shown for comparison. (**D**) Average relative contrast steps toward the background color needed to achieve 85% target detection accuracy for all stimuli. Error bars represent 95% CIs over participants, bar orientation represents stimulus orientation in DKL space. For **C, D**, the length of the scale bar corresponds to the distance from the edge of the hexagon to the outer ring.

The online version of this article includes the following figure supplement(s) for figure 2:

**Figure supplement 1.** Event-related field (ERF) and reaction time results.

## Subjects show hue dependence in reaction times and detection thresholds

As set by the staircase, participants' report accuracy was close to 85% (grating: 83%, $CI_{95\%}$ = [76–87%], colors: 87%, $CI_{95\%}$ = [86–87%], all confidence intervals based on nonparametric bootstraps). However, change-detection performance, as defined by the color contrast of the stimulus change on which the per-color staircases converged, differed across hues ($F(7,203)$ = 82.18, p < 3 × $10^{-16}$). Participants reached 85% accuracy with a change color contrast toward background between 20% ($CI_{95\%}$ = [17%, 23%]) for red-yellow and 37% ($CI_{95\%}$ = [31%, 45%]) for red-blue stimuli (*Figure 2D*). There was one notable outlier: For blue stimuli, the necessary change color contrast toward background was 71% ($CI_{95\%}$ = [66%, 76%]), significantly higher than for other hues ($t(29)$ = 18.01, p < 3 × $10^{-16}$).

On average, participants took 496 ms ($CI_{95\%}$ = [467 ms, 524 ms]) to report the change location on grating stimuli. For color stimuli, they took on average 547 ms ($CI_{95\%}$ = [521 ms, 570 ms]). Across the different color hues, reaction times differed ($F(7,203)$ = 9.26, p < 7 × $10^{-10}$). Subjects were quickest to detect contrast changes on red stimuli (mean = 509 ms, $CI_{95\%}$ = [483 ms, 540 ms]), and slowest for green-yellow stimuli (mean = 562 ms, $CI_{95\%}$ = [535 ms, 587 ms], *Figure 2—figure supplement 1C*).

## Event-related fields are weakest for blue stimuli

Both grating stimuli and equiluminant color disks induced clear visual event-related fields (ERFs) in area V1 (*Figure 2A*), with a shape and timing similar to visually evoked potentials recorded in macaque V1 (Figure 5B in *Rols et al., 2001*). ERFs were *z*-scored relative to the per-trial baseline (see Methods). The ERF differed across stimuli during a prominent component, 57- to 94-ms poststimulus onset (*Figure 2B*, $F(3.48, 101.04) = 30.32$, $p_{GG} < 3 \times 10^{-15}$), which we suspect to be of similar origin as the N70 component recorded in *Rols et al., 2001*. The N70 component was stronger for grating compared to color stimuli (*Figure 2C*) and $t(9) = 6.80$, $p < 2 \times 10^{-7}$). Across stimulus hues, the N70 component was strongest for red (mean = −0.83, $CI_{95\%}$ = [−0.96, −0.71]) and green-yellow stimuli (mean = −0.82, $CI_{95\%}$ = [−0.94, −0.70]), and weakest for blue stimuli (mean = −0.33, $CI_{95\%}$ = [−0.38, −0.29]).

The initial ERF slope is sometimes used to estimate feedforward drive. We extracted the per-participant, per-color N70 initial slope and found significant differences across hues ($F(4.89, 141.68) = 7.53$, $p_{GG} < 4 \times 10^{-6}$). Specifically, it was shallower for blue hues compared to all other hues (all $p_{Holm} < 7 \times 10^{-4}$) except for green and green-blue, while it was not significantly different between all other stimulus hue pairs (all $p_{Holm} > 0.07$, *Figure 2—figure supplement 1A*), demonstrating that stimulus drive (as estimated by ERF slope) was approximately equalized over all hues but blue.

The peak time of the N70 component was significantly later for blue stimuli (mean = 88.6 ms, $CI_{95\%}$ = [84.9 ms, 92.1 ms]) compared to all (all $p_{Holm} < 0.02$) but yellow, green, and green-yellow stimuli, for yellow (mean = 84.4 ms, $CI_{95\%}$ = [81.6 ms, 87.6 ms]) compared to red and red-blue stimuli (all $p_{Holm}$ < 0.03), and fastest for red stimuli (mean = 77.9 ms, $CI_{95\%}$ = [74.5 ms, 81.1 ms]) showing a general pattern of slower N70 peaks for stimuli on the S-(L+M) axis, especially for blue (*Figure 2—figure supplement 1B*).

## Color stimuli induce V1 gamma oscillations measurable in MEG

In addition, color and grating stimuli induced significant visual narrowband gamma-band power increases in early visual cortex (*Figure 3A–D, G, H*). Color stimuli induced gamma-power increases at the subject-specific gamma peak of on average 19% ($CI_{95\%}$ = [14–26%]) relative to the baseline, whereas grating stimuli induced gamma-power increases of 100% ($CI_{95\%}$ = [80–120%], $t_{difference}(29) = 8.32$, $p < 4 \times 10^{-9}$).

Grating- and color-induced gamma source-localized to similar sources, being strongest in areas V1/V2 and extending into parietal and temporal cortex (*Figure 3H*). For both grating and color stimuli, induced gamma-power changes were strongest in areas V1 (grating: 142%; color: 21%), V2 (grating: 131%, color: 19%), V3 (grating: 113%, color: 19%), and V4 (grating: 91%, color: 17%), respectively.

To test for the existence of gamma peaks, we fit the power-change spectra (per participant and per stimulus, averaged over trials) with (1) the sum of two Gaussians and a linear slope, (2) the sum of one Gaussian and a linear slope, and (3) only a linear slope (without any peaks); subsequently, we chose the best-fitting model using adjusted $R^2$ values. Ninety-seven percent of all participants showed a significant gamma peak for at least one color and for the grating stimulus. A second, higher gamma peak was detected in 90% of participants for at least one color stimulus. While induced gamma was five times stronger for grating compared to color stimuli, only 13% of participants showed a second gamma peak when the grating stimulus was shown. For the two peaks often observed with color stimuli, there was no linear relation between the two fitted peak frequencies across colors and participants ($r = 0.06$, $p = 0.58$, *Figure 3F*), indicating that the second peak was unlikely to be a harmonic of the first. As expected due to interindividual variability in skull thickness, cortical dipole orientation, genetic factors, and peak cone sensitivities (*Butler et al., 2019*; *Neitz and Neitz, 2011*; *van Pelt et al., 2012*; *van Pelt et al., 2018*), there was substantial variability in color-induced gamma-peak power and frequency over participants (*Figure 3—figure supplement 2*).

There was a significant difference between the average lower gamma-peak frequencies induced by grating (51.6 Hz, $CI_{95\%}$ = [49.4 Hz, 53.7 Hz]) and color stimuli (45.9 Hz, $CI_{95\%}$ = [42.6 Hz, 49.1 Hz], $p_{difference} < 3 \times 10^{-3}$). Across participants, the gamma-peak frequencies induced by grating and by color stimuli were correlated ($r = 0.64$, $p < 3 \times 10^{-4}$). However, none of the different color pairs showed significantly differing peak frequencies in pairwise comparisons within participants across different colors (*Figure 3—figure supplement 1B*, all $p_{Holm} > 0.36$), also indicating that input drive was roughly

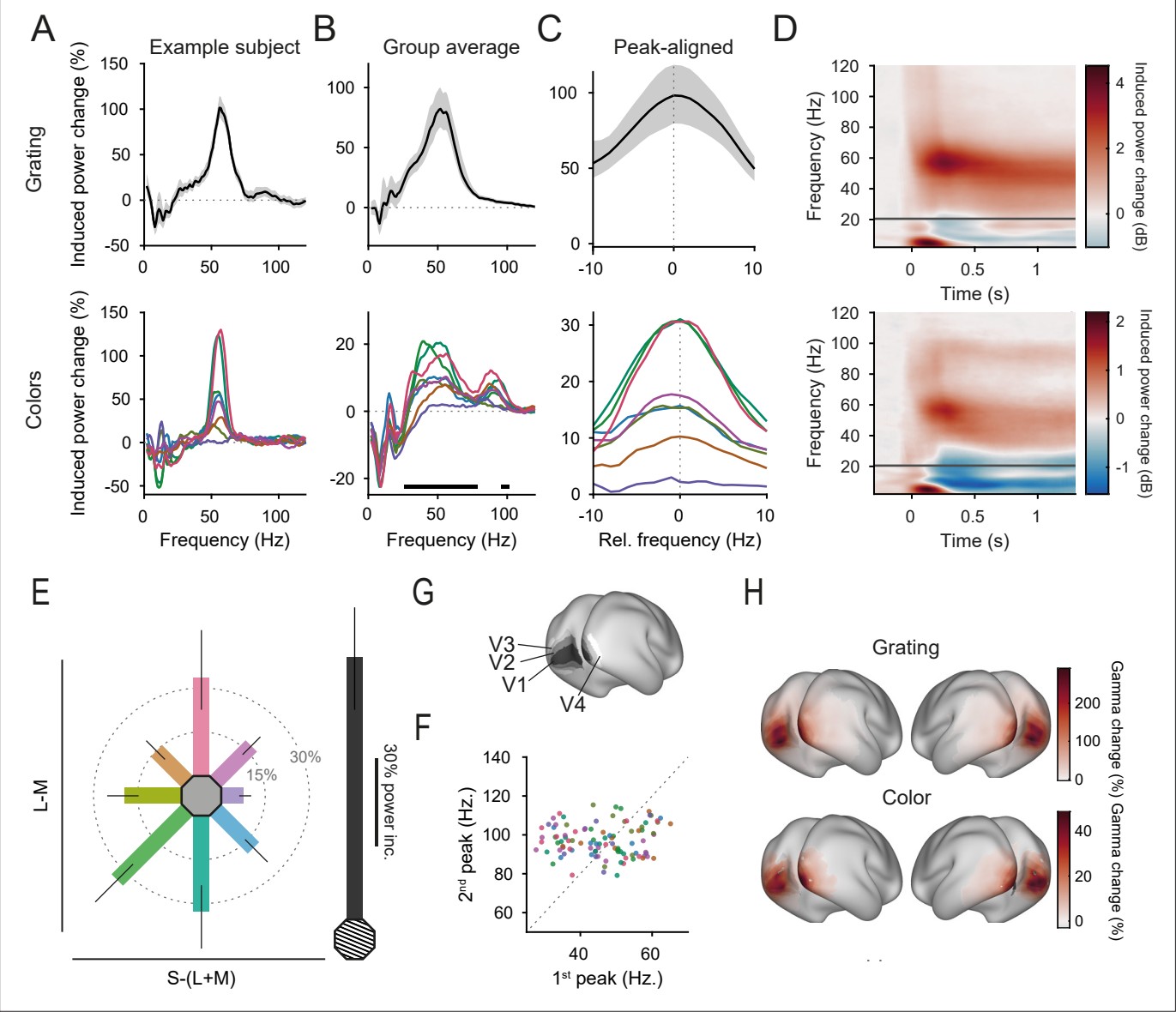

**Figure 3.** Gamma-band responses. (**A**) Stimulus-induced power changes over baseline for one example participant that showed strong gamma-band responses (averaged over V1 dipoles). Top: grating-induced power changes. Error region shows 95% CI over trials. Bottom: color-induced power changes. (**B**) Same as A, but averaged over participants. Error region shows 95% CI over participants. (**C**) Same as B, but peak-aligned before averaging. For A–C, line colors reflect stimulus colors, but have been darkened for readability. (**D**) Average stimulus-induced power change in V1 as a function of time and frequency. Top: for grating stimuli. Bottom: for green stimuli. (**E**) Average stimulus-induced gamma-power change (individual gamma peak ±10 Hz) for all stimuli. Error bars represent 95% CIs over participants, bar orientation represents stimulus orientation in DKL space. In gray, the average grating-induced gamma-power strength is shown for comparison. The length of the scale bar corresponds to the distance from the edge of the hexagon to the outer ring. (**F**) Gamma-peak frequencies of the first and second gamma peak for all participant–color combinations in which a first and a second gamma peak was found. Dot color corresponds to stimulus color. Dotted line indicates the expected frequency relationship, if first and second peak frequencies were harmonics of each other. (**G**) The inflated template brain. Black-to-white shading indicates areas V1, V2, V3, and V4. (**H**) Average stimulus-induced gamma-power change (individual gamma peak ±10 Hz), source projected to all cortical dipoles. Values are significance masked using false discovery rate control (*Benjamini and Yekutieli, 2001*), black-to-white shading indicates areas V1, V2, V3, and V4. All panels show power change 0.3–1.3 s after stimulus onset, relative to baseline.

The online version of this article includes the following figure supplement(s) for figure 3:

**Figure supplement 1.** Spectral measures.

**Figure supplement 2.** Individual spectra.

**Figure supplement 3.** Per-color time–frequency responses.

equalized across colors (*Lewis et al., 2021*; *Ray and Maunsell, 2010*; *Roberts et al., 2013*). The higher peak did not differ in peak frequencies induced by grating (96.5 Hz, $CI_{95\%}$ = [91.8 Hz, 101.3 Hz]) and color stimuli (97.1 Hz, $CI_{95\%}$ = [95.0 Hz, 99.1 Hz], $p_{Difference}$ = 0.68) and showed no significant differences in pairwise comparisons across different colors (*Figure 3—figure supplement 1C,D* all $p_{Holm}$ > 0.37).

## Induced gamma power does not differ between equiluminant red and green stimuli

Crucially, for the red and green stimuli that were chosen to have equal absolute L-M contrast, the induced gamma-power change was not statistically different between the two (*Figure 3E*, red: 30.8%, $CI_{95\%}$ = [20.3%, 43.5%], green: 31.2%, $CI_{95\%}$ = [21.2%, 42.2%], $t(29)$ = 0.135, $p_{Holm}$ = 0.89, $BF_{01}$ = 7.0). The Bayes factor can be interpreted as meaning that the data are 7.0 times more likely to occur under a hypothesis of no differences between red and green stimuli compared to a hypothesis assuming differences.

On the S-(L+M) axis, stimuli with equal absolute cone contrasts showed differing induced gamma power: While yellow stimuli induced a 15.4% gamma-power increase over baseline ($CI_{95\%}$ = [10.7%, 21.9%]), blue stimuli showed significantly smaller, very weak increases of only 2.2% ($CI_{95\%}$ = [0.3%, 4.3%], $t(29)$ = 4.02, $p_{difference}$ < 0.001).

To attempt to control for potential remaining differences in input drive that the DKL normalization missed, we regressed out per participant, per color, the N70 slope and amplitude from the induced gamma power. Results remained equivalent along the L-M axis: The induced gamma-power change residuals were not statistically different between red and green stimuli (red: 8.22, $CI_{95\%}$ = [−0.42, 16.85], green: 12.09, $CI_{95\%}$ = [5.44, 18.75], $t(29)$ = 1.35, $p_{Holm}$ = 1.0, $BF_{01}$ = 3.00).

As we found differences in initial ERF slope especially for blue stimuli, we checked if this was sufficient to explain weaker induced gamma power for blue stimuli. While blue stimuli still showed weaker gamma-power change residuals than yellow stimuli when regressing out changes in N70 slope and amplitude (blue: −11.23, $CI_{95\%}$ = [−16.89, −5.57], yellow: −6.35, $CI_{95\%}$ = [−11.20, −1.50]), this difference did not reach significance ($t(29)$ = 1.65, $p_{Holm}$ = 0.88). This suggests that lower levels of input drive generated by equicontrast blue versus yellow stimuli might explain the weaker gamma oscillations induced by them.

## Performance, ERF, and induced gamma power are related across colors

Higher change-detection performance (defined as lower final staircased target color contrast) was correlated with stronger average V1 dipole moment (i.e., the negative ERF amplitude during the N70 component) across colors, in the time period from 67- to 93-ms poststimulus onset (*Figure 4A*, $r_{max}$ = 0.43, all $p_{Tmax}$ < 0.03, corrected for multiple comparisons across time). Higher change-detection

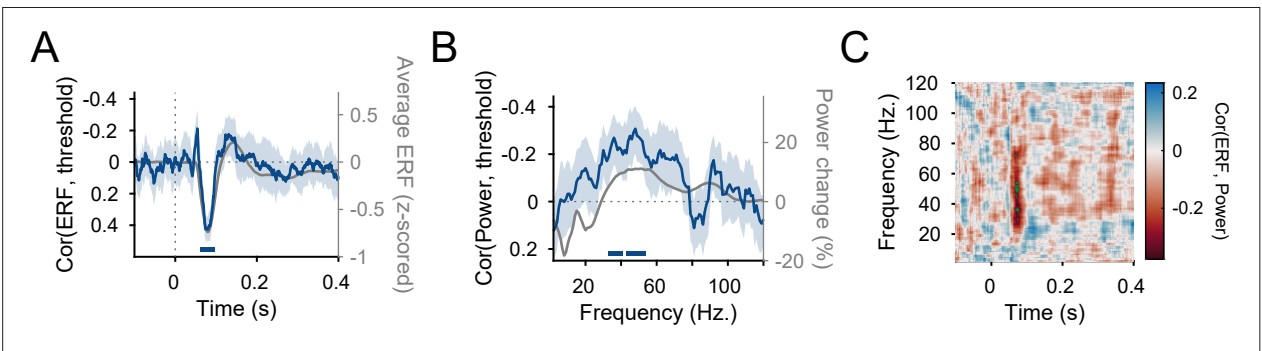

**Figure 4.** Correlations between event-related field (ERF), induced power spectra, and performance. (**A**) Per-timepoint correlation, across colors, between V1 ERF dipole moment (averaged over trials and dipoles) and 85% accuracy threshold. The correlation was first calculated per participant and then averaged over participants. Error bands represent 95% CIs over participants. In gray, the average ERF dipole moment timecourse over color stimuli is shown for comparison. Horizontal blue bar represents significant correlation values (multiple comparison-corrected using $t_{max}$ correction). (**B**) Same as A, but for the stimulus-induced V1 power change spectra instead of the ERFs. (**C**) Correlation, across colors, between V1 ERF dipole moment and V1 stimulus-induced power change, as a function of ERF time and spectral frequencies. The correlation was first calculated per participant and then averaged over participants. Significant correlation values (multiple comparison corrected using $t_{max}$ correction) are indicated by green dots.

performance was also correlated with stronger induced V1 power across colors, for several gamma frequency ranges (*Figure 4B*; at 34–37 Hz: $r_{max} = -0.27$, all $p_{Tmax} \leq 0.03$; at 39–40 Hz: $r_{max} = -0.23$, all $p_{Tmax} < 0.04$; at 44–53 Hz: $r_{max} = -0.31$, all $p_{Tmax} \leq 0.03$, corrected for multiple comparisons across frequency).

In addition, there were significant correlations between the N70 ERF component and induced gamma power: The extracted N70 amplitude was correlated across colors with the induced gamma-power change within participants with on average $r = -0.38$ ($CI_{95\%} = [-0.49, -0.28]$, $p_{Wilcoxon} < 4 \times 10^{-6}$). This correlation was specific to the gamma band and the N70 component: Across colors, there were significant correlation clusters between V1 dipole moment 68- to 79-ms poststimulus onset and induced power between 28–54 and 72 Hz (*Figure 4C*, $r_{max} = -0.30$, $p_{Tmax} < 0.05$, corrected for multiple comparisons across time and frequency).

When perceptual (instead of neuronal) definitions of equiluminance are used, there is substantial between-subject variability in the ratio of relative L- and M-cone contributions to perceived luminance, with a mean ratio of L/M luminance contributions of 1.5–2.3 (*He et al., 2020*). Our perceptual results are consistent with that: We had determined the color-contrast change-detection threshold per color; We used the inverse of this threshold as a metric of color change-detection performance; The ratio of this performance metric between red and green (L divided by M) had an average value of 1.48, with substantial variability over subjects ($CI_{95\%} = [1.33, 1.66]$).

If such variability also affected the neuronal ERF and gamma-power measures reported here, L/M-ratios in color-contrast change-detection thresholds should be correlated with L/M-ratios in ERF amplitude and induced gamma power across participants. This was not the case: Change-detection threshold red/green ratios were neither correlated with ERF N70 amplitude red/green ratios ($\rho = 0.09$, $p = 0.65$, $BF_{01} = 3.99$), nor with induced gamma-power red/green ratios ($\rho = -0.17$, $p = 0.38$, $BF_{01} = 3.04$).

## Discussion

In summary, uniform colored stimuli induced gamma oscillations in humans that were measurable in MEG recordings, confirming a previous study (*Perry et al., 2020*). When red-green color hues were chosen to equalize cone contrast on the L-M axis (and thereby to ideally equalize input drive), the induced V1 gamma power did not differ between red and green hues. Additionally, pure activation of the S-(L+M) axis induced weaker gamma power, especially for color hues activating primarily the S-cone.

This curious asymmetry on the S-(L+M) axis could also be seen in smaller amplitudes of early ERF components (N70) in area V1 and in worse color-contrast change-detection performance for high-S (blue) stimuli. Accordingly, across hues, N70 amplitude and gamma power were correlated with color-contrast change-detection performance. The fact that controlling for N70 amplitude and slope strongly diminished the recorded differences in induced gamma power between S+ and S− stimuli supports the idea that the recorded differences in induced gamma power over the S-(L+M) axis might be due to pure S+ stimuli generating weaker input drive to V1 compared to DKL-equicontrast S− stimuli, even when cone contrasts are equalized.

Concerning off-axis colors (red-blue, green-blue, green-yellow, and red-yellow), we found stronger gamma power and ERF N70 responses to stimuli along the green-yellow/red-blue axis (which has been called lime-magenta in previous studies) compared to stimuli along the red-yellow/green-blue axis (orange-cyan). In human studies varying color contrast along these axes, lime-magenta has also been found to induce stronger fMRI responses (*Goddard et al., 2010*; but see *Lafer-Sousa et al., 2012*), and psychophysical work has proposed a cortical color channel along this axis (*Danilova and Mollon, 2010*; but see *Witzel and Gegenfurtner, 2013*).

In our data, the color-induced gamma oscillation clearly localized to early visual cortex, was narrow-banded and showed (for 90% of subjects) a second spectral peak around 80–100 Hz, which did not seem to be a harmonic of the main induced peak and which was rare for grating stimuli. As the induced gamma-power change source-localized as a continuous source over the early visual cortex with a peak in V1, our data cannot differentiate between a strong gamma source in V1 that shows source smearing over neighboring areas and, alternatively, several gamma sources in V1–V4 that show monotonous decreases in power with distance from V1.

By designing our stimuli to drive the LGN equally, we could test gamma responses of the early visual cortex for color stimuli of approximately equal V1 input strength on the cardinal DKL axes. While most previous studies of gamma responses to colored stimuli were limited to small samples of non-human primates or human patients, recruiting a bigger sample of human participants allowed us to generalize findings of color gamma responses to the population level (*Fries and Maris, 2022*).

When comparing our results to previous reports based on intracranial recordings, it is important to note that the MEG signal does not represent a simple average over all population activity underneath the MEG sensor, but is influenced by the location and orientation of the underlying neuronal dipoles and is most sensitive to synchronized neuronal activity (*Baillet, 2017*; *Murakami and Okada, 2006*). Thereby, the total power reported here does not correspond to the total power of all oscillatory neuronal activity in V1. Additionally, activity sources that are of unequal frequency or phase are likely not fully captured in MEG recordings. On the other hand, intracortical recordings will usually not sample all of area V1 equally, but will emphasize the color tuning of neuronal populations close to the electrodes.

The localization accuracy of source-localized MEG is limited by subject movement, uncertainties in head tracking and source spread. However, with careful head stabilization and exclusion of subjects showing excessive head movements (as practiced here), spatial resolution can be brought down to between 0.45 and 7 mm, depending on dipole location (*Nasiotis et al., 2017*). Additionally, the high similarity between early (source-localized) V1 ERF components recorded here and intracortical V1 ERP components recorded in macaque (*Rols et al., 2001*) suggests that our V1-localized activity in MEG captures the underlying intracortical voltage signal in V1.

## Gamma responses to colored stimuli

In several previous studies in non-human and human primates analyzing early visual cortex gamma-band responses to colored stimuli, a red effect, that is stronger induced gamma power for red versus non-red colors, was reported. In most of these studies, the presented colors were not equiluminant and did not have equal color contrasts to the background.

In an initial report on recordings in anesthetized macaque V1 and V4, *Rols et al., 2001* first described stronger gamma power for a red stimulus compared to a green and a yellow one (heteroluminant, all presented on a gray background). Similar to our study, color-induced gamma oscillations were significantly stronger in V1 than in V4 in their recordings. *Shirhatti and Ray, 2018*, recording in awake macaque V1, presented fullscreen, heteroluminant colors after a gray intertrial background and found strongest gamma power for red and blue hues. When colors were transformed to coordinates based on L- and M-cone activation, induced gamma power was found to increase with cone contrast modulations toward both axis directions in one of the three monkeys tested.

Using natural stimuli, *Brunet et al., 2015* presented 65 grayscale and color images to two macaque monkeys and found that, in ECoG recordings over awake macaque areas V1 and V4, the strongest gamma power was induced by images of a colored orange on a gray background. In another study of macaque V1 responses to natural stimuli, induced gamma power increased with both negative and positive L-M cone contrast (*Peter et al., 2021*).

In intracortical recordings from macaque V1 under visual stimulation with a peripheral colored disk, *Peter et al., 2019* found strongest gamma responses to pink and red stimuli. When colors were chosen from an equiluminance plane in DKL space for a control analysis in one monkey, they still found slightly stronger gamma power for red than for green stimuli, even if red and green stimuli were of the same color contrast. The same asymmetry on the S-(L+M) axis as reported here was found, in the sense that pure blue stimuli induced very weak gamma power. In addition, they explored the effects of prestimulus and background adaptation and found that red induced the strongest gamma power only when presented on a gray background, but not when presented on a black background. Presenting a series of different stimulus and background combinations and summarizing results in a self-defined cone adaptation model, they showed that red and green stimuli induce equal gamma power when the background does not adapt M- and L-cones, and that, over all stimulus/background combinations, M-cone-activating backgrounds boost L-cone-stimulus-driven gamma oscillations the strongest.

In recordings from ECoG electrodes implanted over areas V1–V4 in 10 epilepsy patients, *Bartoli et al., 2019* found strongest gamma-power responses for full-screen red/orange stimuli following a gray intertrial interval background. The presented colors were equiluminant and equicontrast in

CIELAB, a color coordinate system based on perceptual similarity. In ECoG and intracranial EEG recordings from a second set of seven epilepsy patients using the same task and stimuli (*Li et al., 2022*), strongest gamma-power responses were found for red/orange and green stimuli. Color gamma responses reported in both studies were of similar strength (expressed in percent change from baseline) as in our MEG study.

Oscillatory neuronal responses to full-screen colored stimuli have also been reported in MEG recordings from 20 participants (*Perry et al., 2020*). When full-screen red, green, purple, and blue colors were shown after a gray background or in alteration, red stimuli induced the strongest power in several oscillatory bands above 30 Hz.

In total, all previous studies not explicitly controlling for cone contrast found stronger gamma-power responses for red (and sometimes blue) stimuli than for other colors. When cone contrast was controlled for in two macaque V1 studies, either through the use of DKL coordinates or through a self-defined cone adaption model, the difference between red and green stimuli either decreased (*Peter et al., 2019*; *Shirhatti and Ray, 2018*) or, when adaptation was also taken into account, disappeared (*Peter et al., 2019*).

Interestingly, a similar case has been found before in other measures of V1 activation: Macaque V1 glucose uptake, intrinsic signal imaging and 2-photon calcium imaging found strongest responses for the employed red and blue stimuli (*Garg et al., 2019*; *Li et al., 2021*; *Liu et al., 2020*; *Tootell et al., 1988*; *Xiao et al., 2007*). These so-called end-spectral biases were likely at least partially an effect of the respective stimulus presentation and color coordinate systems, insofar as their red and blue colors induced maximal cone contrasts in the L-M pathway – highest L-M excitation for their red stimuli, and highest L excitation for their blue stimuli (*Li et al., 2021*; *Liu et al., 2020*; *Mollon, 2009*; *Valverde Salzmann et al., 2012*).

## Lower responses for short-wavelength stimuli

In our data and in a previous report (*Peter et al., 2019*), induced gamma power was not equal between equicontrast stimuli on the S-(L+M) axis: Blue stimuli showed no or very weak gamma-power increases. Similarly, we found that change-detection performance was worse and ERF amplitudes were smaller. This asymmetry on the S-(L+M) axis might be driven by underlying physiology: Compared to L- and M-cones, S-cones are sparser (*DeMonasterio et al., 1981*) and show slower light responses, higher noise levels, and higher activation thresholds (*Baudin et al., 2019*; *Cole et al., 1993*; *Lee et al., 2009*). Additionally, S-(L+M)-sensitive neurons are sparser than L-M-sensitive ones in the LGN (*Derrington et al., 1984*) and V1 (*Li et al., 2021*). Within V1, neurons sensitive to S-cone inputs reside in different cortical layers and respond later than other color-sensitive neurons (*Cottaris and De Valois, 1998*). Likely, S-cone-induced signals are comparatively weak within LGN and are amplified and transformed in V1 (*De Valois et al., 2000*; *Mullen et al., 2008*; *Xiao, 2014*).

Contrary to the unified pathway for L-M activation, stimuli high and low on the S-(L+M) axis (S+ and S−) each target different cell populations in the LGN, and different cortical layers within V1 (*Chatterjee and Callaway, 2003*; *De Valois et al., 2000*), whereby the S+ pathway shows higher LGN neuron and V1 afferent input numbers (*Chatterjee and Callaway, 2003*). Other metrics of V1 activation, such as ERPs/ERFs, reveal that these more numerous S+ inputs result in a weaker evoked potential that also shows a longer latency (our data; *Nunez et al., 2021*). The origin of this dissociation might lie in different input timing or less cortical amplification, but remains unclear so far. Interestingly, our results suggest that cortical gamma is more closely related to the processes reflected in the ERP/ERF: Stimuli inducing stronger ERF induced stronger gamma; and controlling for ERF-based measures of input drives abolished differences between S+ and S− stimuli in our data.

The source localization to V1, the dependence on input drive (L-M contrast), and the size dependence (*Peter et al., 2019*) are features that gamma responses induced by L-M input share with gamma responses induced by luminance contrast. This points to common principles in the generation of gamma for both chromatic and achromatic stimuli. Yet, the fact that colored disks as compared to gratings often induced a second, higher-frequency gamma peak suggests differences in underlying circuits and/or dynamics.

## A potential source of V1 gamma selectivity across color hues

In sum, color-induced gamma responses in area V1 match LGN-to-V1 input strength in three measured aspects: (1) along the L-M axis, stimuli driving LGN to an equal degree induce gamma oscillations that do not differ measurably in strength across the population; (2) gamma oscillations are strongly diminished for S-cone-driving stimuli, which are encoded by smaller populations of LGN neurons and which provide weaker excitatory input into V1, compared to L-M-cone-driving stimuli; (3) the amplitude of the early, input-driven N70 component is correlated with later induced gamma power over stimuli. Previous reports of high color specificity of V1 gamma oscillations might therefore be explained by a positive relationship between the strength of feedforward input into V1 and the power of V1 gamma oscillations.

## Materials and methods

### Participants

A sample size of 30 was chosen based on current standards in the MEG field. Thirty participants were recruited from the general public, 18 of them female. They were between 18 and 36 years old (average: 26 years). As participants were recruited via public advertisements, most of them had not participated in vision-science experiments before. They were screened to be free of metal implants, did not take medication during the study period except for contraceptives, had never been diagnosed with any neurological or psychological disorder, had normal or corrected-to-normal vision and did not show red-green color vision deficiencies. The screening criterion for red-green deficiencies was correct answer to all of a nine-plate subset of the 38-plate Ishihara test (*Ishihara, 1979*). A further five subjects were recruited but excluded due to equipment malfunctions or excessive movements (>5 mm) during the experiment. The study was approved by the ethics committee of the medical faculty of the Goethe University Frankfurt (Resolution E 36/18).

### Paradigm

Participants were seated in a dark, magnetically shielded, sound-dampened room. Stimuli were shown on a backprojection screen with a distance of 58 cm to their eyes using a Propixx projector (resolution: 960 × 520 px, refresh rate: 480 Hz), controlled with Psychtoolbox-3 (*Kleiner et al., 2007*). Eye position and pupil size were recorded using an infrared eye tracking system (EyeLink 1000). Before the experiment, participants were trained to minimize saccades and blinks during the baseline and trial periods.

The 540 total trials of the experiment were split into two blocks, with 240 color (30 per color) and 30 grating trials shown during each block. Between blocks, participants were given a short break of no longer than 5 min. Within each block, the trial order was chosen randomly for each participant.

A gray background and a dark-gray fixation spot was shown throughout the experiment. Each trial was initiated once the participant fixated the central fixation spot. After a baseline of 1.2 s, the stimulus was shown as a central 10 dva diameter disk with broadly antialiased, smoothed edges. After a per-trial randomly chosen period of 0.3–2 s (randomized, Cauchy distributed with $x_0 = 1.65$ s, FWHM = 0.2 s), a to-be-detected local color change (a circular, Gaussian-shaped color step toward the background color, 3.7 dva diameter) was shown at a random position on the stimulus. Participants were instructed to speedily report the left–right position of the target relative to the fixation dot using a button press (left thumb to indicate left, right thumb to indicate right). The relative transparency of the target to the stimulus was QUEST staircased (*Kleiner et al., 2007*; *Watson and Pelli, 1983*) for each stimulus condition and participant to 85% correct responses. After a button press or after 1 s, the trial was terminated and, if the correct button had been pressed, a smiley was shown for 0.5 s. A random 5% of trials were target-free catch trials.

### Stimuli

Participants were shown equiluminant colors on an isoluminant background as well as a grating stimulus. To define equiluminance, the projector base-color spectral power distributions were measured using an Ocean Insight FLAME-T spectrometer and used to define per-color cone excitation values using human cone spectral sensitivities for 10 dva diameter stimuli (*Stockman and Sharpe, 2000*; *Stockman et al., 1999*). Then, cone excitation values of the neutral gray background and the stimuli themselves were used to define per-color coordinates in DKL color space, an opponent modulation

space originally developed to describe responses of LGN neurons to color stimuli (*Derrington et al., 1984*; *Westland et al., 2012*). $k_{Lum}$, $k_{L-M}$, and $k_{S-L+M}$ were chosen such that on-axis stimuli with unit pooled cone contrast gave unit coordinates in the three DKL axes (*Brainard, 1996*). The eight color stimuli were chosen from the DKL equiluminance plane to a common, gray background and were chosen to be distributed along an ellipse filling the projector color gamut with equal distances between the color coordinates along the ellipse. For the grating condition, a 1.5 cycle/dva antialiased square wave grating rotated 22.5° clockwise from vertical was used.

## MEG recording

Data were recorded using a CTF Systems 275 axial gradiometer MEG system, low-pass filtered (300 Hz) and digitized (1200 Hz). Initial head position was set to minimize distance between the occipital pole and posterior MEG gradiometers. Head position was monitored continuously throughout the experiment, and experiment sessions were aborted and excluded from analysis when participants moved their head more than 5 mm from its initial position. Flexible head fixation using memory foam cushions and medical tape was used throughout the experiment.

## Data analysis

Data were analyzed using custom Matlab and R code and the FieldTrip toolbox (*Oostenveld et al., 2011*). Line noise was removed using DFT filters at 50 Hz and its higher harmonics. The recording was cut into trials from -1 s to 1.3 s relative to stimulus onset. Trials with stimulus changes before 1.3 s after stimulus onset, trials with missing/early responses, and catch trials were removed. Blink, muscle, and SQUID-jump artifacts were detected using a semiautomated artifact detection process. For further analysis, trials were segmented into epochs as detailed below, and analyses were only run for epochs devoid of artifacts. The described trial removal and artifact epoch rejection procedures rejected 19% of all trials.

## Source localization

Analyses localizing power at the participant-specific gamma peak used dynamic imaging of coherent sources beamforming (*Gross et al., 2001*). For other analyses, linearly constrained minimum variance (LCMV) beamforming (*Van Veen et al., 1997*) was used to generate virtual dipole timecourses for all analyzed dipoles. For both beamformers, the covariance matrix was not regularized ($\lambda$ = 0%), and dipoles were placed at all vertices of both hemispheres of the 32 k HCP-MMP1.0 atlas (*Glasser et al., 2016a*). The atlas was registered to subject-specific T1- and T2-weighted MRI scans (T1: 1 mm$^3$ MPRAGE with TR = 2530 ms, TE = 2.27 ms; T2: 1 mm$^3$ TSE with TR = 1500 ms and TE = 356 ms, acquired on a 3T Siemens Magnetom Prisma) using Freesurfer (*Fischl, 2012*) and HCP Workbench (*Glasser et al., 2016b*). Area-specific analyses (e.g., analyses focusing on area V1) averaged results over all dipoles within the specific area of the atlas (1618 dipoles within V1). For the calculation of full spectra, to reduce computational demand, a reduced MMP1.0 atlas subsampled to 6.25% of all 64,000 dipole positions was used, which corresponds to 106 dipoles in area V1. Virtual dipole timecourses were computed by multiplying the sensor-level data with the LCMV filters.

## Behavioral, ERF, and spectral analysis

Per-stimulus change-detection performance was defined as the target-change color contrast the per-stimulus staircase had converged to, averaged over the last 10 presentations of each stimulus.

To analyze source-localized ERFs, we low-pass filtered the V1 virtual dipole timecourses using an acausal Gaussian filter kernel (−6 dB at 80 Hz), cut them to −0.2 to 0.6 s relative to stimulus onset and z-scored them relative to the per-trial baseline. Then, we averaged over all V1 dipoles and trials of the same stimulus type within participants. Because the ERF polarity is dependent on dipole orientation and not consistent throughout V1, we flipped (per participant and dipole, yet identical for all stimuli) virtual dipole ERFs that were positive-going during the N70 component (55- to 95-ms poststimulus onset) before averaging. ERF peak time of the first negative-going (N70) component was defined as the timepoint between 50- and 110-ms poststimulus onset at which the per-participant, per-stimulus average ERF reached its minimum. The slope of this component was taken from the time period 2–12 ms before this peak. Per-trial N70 amplitude was then defined as the average V1 dipole amplitude in a window of ±10 ms around the per-stimulus N70 peak time.

For spectral analyses, virtual dipole timecourses were cut into baseline ($-1.0$ to 0 s relative to stimulus onset) and stimulus (0.3 to 1.3 s after stimulus onset) periods. Then, they were segmented into 50% overlapping time epochs of 500 ms (for frequencies $\leq$20 Hz) or 333 ms length (for frequencies >20 Hz), demeaned and detrended. They were then Hann-tapered (for frequencies $\leq$20 Hz) or Slepian-window multitapered using three tapers to achieve $\pm$3 Hz smoothing (for frequencies >20 Hz), zero-padded to 1 s and Fourier transformed. Those Fourier spectra were used to obtain source-level power spectra. Spectra of stimulus-induced power change were computed relative to the per-stimulus averaged baseline spectra and then averaged over the selected dipoles. For time–frequency analyses, the same pipeline with epochs centered on each timepoint of the virtual dipole timecourse (sampling rate = 1200 Hz) and zero-padding to 4 s was run. To define per-trial power change values, we extracted the frequency bin centered on the per-participant, per-stimulus gamma-peak frequency.

To determine per-stimulus, per-participant gamma-peak frequencies, the trial-average spectra of stimulus-induced power change from 15 to 120 Hz were fit with three model alternatives: (1) A first-order polynomial; (2) The sum of a first-order polynomial and a Gaussian; (3) The sum of a first-order polynomial and two Gaussians. Spectral bins were weighted according to their scaled power change, and both Gaussians were set to have a minimum amplitude of 5% and a standard deviation between 1.5 and 10 Hz. For alternative (2), the single Gaussian was set to have a mean above 21; for alternative (3), one Gaussian was set to have a mean above 21 Hz, and the second Gaussian was set to have a mean above 61 Hz. The model giving the highest adjusted $R^2$ was chosen, and gamma-peak frequencies of the found peaks were extracted. If no gamma-peak frequency was found for a given participant–stimulus pair, analyses that required a gamma-peak frequency (DICS beamforming and extracting power at the gamma peak) were computed using the average gamma-peak frequency over stimuli within that participant. These participant–stimulus combinations were excluded from analyses explicitly reporting peak frequency values.

To correlate per-color average ERFs and induced spectra with each other and with behavioral performance, we averaged per-stimulus V1 ERFs and per-stimulus V1 change spectra over trials within each participant. Then, for each timepoint/frequency bin, the per-stimulus value was correlated to the per-stimulus change-detection performance. The results were corrected for multiple comparisons over timepoints/frequency bins using $t_{max}$ multiple-comparison correction (**Blair et al., 1994**).

## Statistical analysis

Reported tests were two tailed, alpha was set to $\alpha$ = 0.05. To calculate evidence toward the null hypothesis for pairwise comparisons and Spearman's rank correlations, we computed Bayes factors of $H_0$ over $H_1$ using a prior scale of one (**Rouder et al., 2009**).

To test for significant differences across colors in measures of interest (ERF amplitudes, induced spectral power changes, behavioral performance, reaction times), we averaged these measures per stimulus within participants and then performed RMANOVAs across stimuli. The resulting $F$ and p values were Greenhouse–Geisser corrected where appropriate. Pairwise comparisons were then computed using $t$-tests, which were Bonferroni–Holm corrected for multiple comparisons.

## Acknowledgements

We thank the reviewers for constructive input, and Karl R Gegenfurtner, Jarrod R Dowdall, Cem Uran, and Joscha T Schmiedt for insightful discussions about color spaces. We also thank the ESI workshop and IT teams as well as the BIC MEG and MRI infrastructure groups for their support.

## Additional information

### Competing interests

Pascal Fries: has a patent on thin-film electrodes (US20170181707A1) and is beneficiary of a respective license contract on thin-film electrodes with Blackrock Microsystems LLC (Salt Lake City, UT), is member of the Advisory Board of CorTec GmbH (Freiburg, Germany), and managing director of Brain Science GmbH (Frankfurt am Main, Germany). The authors declare no further competing interests. The other authors declare that no competing interests exist.

## Funding

| Funder | Grant reference number | Author |
|---|---|---|
| Deutsche Forschungsgemeinschaft | FOR 1847 FR2557/2-1 | Pascal Fries |
| Deutsche Forschungsgemeinschaft | FR2557/5-1-CORNET | Pascal Fries |
| Deutsche Forschungsgemeinschaft | FR2557/7-1 DualStreams | Pascal Fries |
| European Union | FP7-604102-HBP | Pascal Fries |
| International Max Planck Research School for Neural Circuits | open access funding | Benjamin J Stauch |

The funders had no role in study design, data collection, and interpretation, or the decision to submit the work for publication.

## Author contributions

Benjamin J Stauch, Conceptualization, Formal analysis, Investigation, Software, Visualization, Writing – original draft, Writing – review and editing; Alina Peter, Conceptualization, Writing – review and editing; Isabelle Ehrlich, Investigation, Writing – review and editing; Zora Nolte, Formal analysis, Software; Pascal Fries, Conceptualization, Funding acquisition, Supervision, Writing – review and editing

## Author ORCIDs

Benjamin J Stauch http://orcid.org/0000-0002-4484-813X
Alina Peter http://orcid.org/0000-0001-8497-6235
Pascal Fries http://orcid.org/0000-0002-4270-1468

## Ethics

Subjects gave written informed consent. The study was approved by the ethics committee of the medical faculty of the Goethe University Frankfurt (Resolution E 36/18).

## Decision letter and Author response

Decision letter https://doi.org/10.7554/eLife.75897.sa1
Author response https://doi.org/10.7554/eLife.75897.sa2

# Additional files

## Supplementary files

- MDAR checklist
- Transparent reporting form

## Data availability

Custom preprocessing code, per-trial data and code for statistical analyses are available at https://doi.org/10.5281/zenodo.5578940.

The following dataset was generated:

| Author(s) | Year | Dataset title | Dataset URL | Database and Identifier |
|---|---|---|---|---|
| Stauch BJ, Peter A, Ehrlich I, Nolte Z, Fries P | 2022 | Dataset for Human visual gamma for color stimuli | https://doi.org/10.5281/zenodo.5578940 | Zenodo, 10.5281/zenodo.5578940 |

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
