## [Editor Report]

Previous work has shown that presentation of uniform fields of color can induce strong gamma rhythms in visual cortex of monkeys and humans, particularly red fields. However, prior work has rarely considered the effectiveness of the colored visual stimulus at driving the input to visual cortex. This study uses MEG measurements from human visual cortex and shows that this preference for red is reduced when colored stimuli are likely to drive similar levels of input to visual cortex.

---

## [Decision Letter]

**Decision letter after peer review:**

Thank you for submitting your article "Human visual gamma for color stimuli: When LGN drive is equalized, red is not special" for consideration by *eLife*. Your article has been reviewed by 3 peer reviewers, and the evaluation has been overseen by Laura Colgin as the Senior Editor. The following individual involved in review of your submission has agreed to reveal their identity: Christopher I Moore (Reviewer #1).

*Reviewer #1 (Recommendations for the authors):*

This is a well-conducted study and speaks to an interesting finding in an important topic, whether ethological validity causes co-variation in gamma above and beyond the already present ethological differences present in systemic stimulus sensitivity.

I like the fact that while this finding (seeing red = ethnologically valid = more gamma) seems to favor views the PI has argued for, the paper comes to a much simpler and more mechanistic conclusion. In short, it's good science.

I think they missed a key logical point of analysis, in failing to dive into ERF <-- gamma relationships. In contrast to the modeled assumption that they have succeeded in color matching to create matched LGN output, the ERF and its distinct features are metrics of afferent drive in their own data. And, their data seem to suggest these two variables are not tightly correlated, so ate very least it is a topic that needs treatment and clarity as discussed below.

I. I would like to see a more precise treatment of ERF and gamma power. The initial slope of the ERF should, by typical convention, correlate strongly with input strength, and the peak should similarly be a predictor of such drive, albeit a weaker one. Figure 4C looks good, but I'm totally confused about what this is showing. If drive = gamma in color space, then these ERF features and gamma power should (by Occham's sledgehammer…) be correlated. I invoke the sledgehammer not the razor because I could easily be wrong, but if you could unpack this relationship convincingly, this would be a far stronger foundation for the 'equalized for drive, gamma doesn't change across colors' argument…(see also IIB below)…

…and, in my own squinting, there is a difference (~25%) in the evoked dipole amplitudes for the vertically aligned opponent pairs of red- and green (along the L-M axis Figure 2C) on which much hinges in this paper, but no difference in gamma power for these pairs. How is that possible? This logic doesn't support the main prediction that drive matched differences = matched gamma…Again, I'm happy to be wrong, but I would to see this analyzed and explained intuitively.

II. As indicated above, the paper rests on accurate modeling of human LGN recruitment, based in fact on human cone recruitment. However, the exact details of how such matching was obtained were rapidly discussed-this technical detail is much more than just a detail in a study on color matching: I am not against the logic nor do I know of a flaw, but it's the hinge of the paper and is dealt with glancingly.

A. Some discussion of model limitations.

B. Why it's valid to assume LGN matching has been achieved using data from the periphery: To buy knowledge, nobody has ever recorded single units in human LGN with these color stimuli…in contrast, the ERF is 'in their hands' and could be directly related (or not) to gamma and to the color matching predictions of their model.

*Reviewer #2 (Recommendations for the authors):*

The major strengths of this study are the use of MEG measurements to obtain spatially resolved estimates of gamma rhythms from a large(ish) sample of human participants, during presentation of stimuli that are generally well matched for cone contrast. Responses were obtained using a 10deg diameter uniform field presented in and around the centre of gaze. The authors find that stimuli with equivalent cone contrast in L–M axis generated equivalent gamma – ie. that 'red' (+L–M) stimuli do not generate stronger responses than 'green (–L+M). The MEG measurements are carefully made and participants performed a decrement-detection task away from the centre of gaze (but within the stimulus), allowing measurements of perceptual performance and in addition controlling attention.

There are a number of additional observations that make clear that the color and contrast of stimuli are important in understanding gamma. Psychophysical performance was worst for stimuli modulated along the +S–(L+M) direction, and these directions also evoked weakest evoked potentials and induced gamma. There also appear to be additional physiological asymmetries along non-cardinal color directions (e.g. Figure 2C, Figure 3E). The asymmetries between non-cardinal stimuli may parallel those seen in other physiological and perceptual studies and could be drawn out (e.g. Danilova and Mollon, Journal of Vision 2010; Goddard et al., Journal of Vision 2010; Lafer-Sousa et al., JOSA 2012). Similarly, the asymmetry between +S and -S modulation is striking and need better explanation within the model (that thalamic input strength predicts gamma strength) given that +S inputs to cortex appear to be, if anything, stronger than –S inputs (e.g. DeValois et al. PNAS 2000).

My only real concern is that the authors use a precomputed DKL color space for all observers. The problem with this approach is that the isoluminant plane of DKL color space is predicated on a particular balance of L- and M-cones to Vlambda, and individuals can show substantial variability of the angle of the isoluminant plane in DKL space (e.g. He, Cruz and Eskew, Journal of Vision 2020). There is a non-negligible chance that all the responses to colored stimuli may therefore be predicted by projection of the stimuli onto each individual's idiosyncratic Vlambda (that is, the residual luminance contrast in the stimulus). While this would be exhaustive to assess in the MEG measurements, it may be possible to assess perceptually as in the He paper above or by similar methods. Regardless, the authors should consider the implications – this is important because, for example, it may suggest that important of signals from magnocellular pathway, which are thought to be important for Vlambda.

Please show sign/scale on axes in Figure 2C/D and similar for legibility

Please report the cone contrast values of the stimuli that were presented

*Reviewer #3 (Recommendations for the authors):*

This is an interesting article studying human color perception using MEG. The specific aim was to study differences in color perception related to different S-, M-, and L-cone excitation levels and especially whether red color is perceived differentially to other colors. To my knowledge, this is the first study of its kind and as such very interesting. The methods are excellent, and manuscript is well written as expected this manuscript coming from this lab. However, illustrations of the results is not optimal and could be enhanced.

The results presented in the manuscript are very interesting, but not presented comprehensively to evaluate the validity of the results. The main results of the manuscript are that the gamma-band responses to stimuli with absolute L-M contrast i.e. green and red stimuli do not differ, but they differ for stimuli on the S-(L+M) (blue vs red-green) axis and gamma-band responses for blue stimuli are smaller. These data are presented in figure 3, but in its current form, these results are not well conveyed by the figure. The main results are illustrated in figures 3BC, which show the average waveforms for grating and for different color stimuli. While there are confidence limits for the gamma-band responses for the grating stimuli, there are no confidence limits for the responses to different color stimuli. Therefore, the main results of the similarities / differences between the responses to different colors can't be evaluated based on the figure and hence confidence limits should be added to these data. It is also not clear from the figure legend, from which time-window data is averaged for the waveforms.

The time-resolved profile of gamma-power changes are illustrated in Figure 3D. This figure would a perfect place to illustrate the main results. However, of all color stimuli, these TFRs are shown only for the green stimuli, not for the red-green differences nor for blue stimuli for which responses were smaller. Why these TFRs are not showed for all color stimuli and for their differences?

All main results are now only shown for data averaged over V1. Taken that V4 that is traditionally associated with color processing, the same results could also be shown for this region.

The authors are in results referring to a second higher gamma peak that was detected in 90% of participants for color but not for grating stimuli. How was the within-subject statistics performed to evaluate the presence of this peak?

The authors discuss the localization accuracy of source-localized MEG. In addition to rough approximations, the authors could also evaluate the specific source localization accuracy of their method in their specific subject population by using simulated data together with realistic forward and inverse modelling. Taken that V1 as a medial structure might not be well source-modelled, this could give important insight on the factors influencing the results.

The correlation of the gamma-power but not ERF with behavioral performance is interesting. The authors might want to elaborate and refer to Julku et al., 2021 Sci. Rep. which found a correlation between oscillations and EDRTs values but not with EFRs and EDTRs values.

---

## [Author Response]

Reviewer #1 (Recommendations for the authors):This is a well-conducted study and speaks to an interesting finding in an important topic, whether ethological validity causes co-variation in gamma above and beyond the already present ethological differences present in systemic stimulus sensitivity.I like the fact that while this finding (seeing red = ethnologically valid = more gamma) seems to favor views the π has argued for, the paper comes to a much simpler and more mechanistic conclusion. In short, it's good science.I think they missed a key logical point of analysis, in failing to dive into ERF <-- gamma relationships. In contrast to the modeled assumption that they have succeeded in color matching to create matched LGN output, the ERF and its distinct features are metrics of afferent drive in their own data. And, their data seem to suggest these two variables are not tightly correlated, so ate very least it is a topic that needs treatment and clarity as discussed below.

Further ERF analyses are detailed below.

I. I would like to see a more precise treatment of ERF and gamma power. The initial slope of the ERF should, by typical convention, correlate strongly with input strength, and the peak should similarly be a predictor of such drive, albeit a weaker one. Figure 4C looks good, but I'm totally confused about what this is showing. If drive = gamma in color space, then these ERF features and gamma power should (by Occham's sledgehammer…) be correlated. I invoke the sledgehammer not the razor because I could easily be wrong, but if you could unpack this relationship convincingly, this would be a far stronger foundation for the 'equalized for drive, gamma doesn't change across colors' argument…(see also IIB below)……and, in my own squinting, there is a difference (~25%) in the evoked dipole amplitudes for the vertically aligned opponent pairs of red- and green (along the L-M axis Figure 2C) on which much hinges in this paper, but no difference in gamma power for these pairs. How is that possible? This logic doesn't support the main prediction that drive matched differences = matched gamma…Again, I'm happy to be wrong, but I would to see this analyzed and explained intuitively.

As suggested by the reviewer, we have delved deeper into ERF analyses. Firstly, we overhauled our ERF analysis to extract per-color ERF shape measures (such as timing and slope), added them as panels A and B in Figure 2—figure supplement 1:

We have revised the results to report those analyses:

“The initial ERF slope is sometimes used to estimate feedforward drive. We extracted the per-participant, per-color N70 initial slope and found significant differences over hues (F(4.89, 141.68) = 7.53, p_GG_ < 4*10^–6^). Specifically, it was shallower for blue hues compared to all other hues except for green and green-blue (all p_Holm_ < 7*10^–4^), while it was not significantly different between all other stimulus hue pairs (all p_Holm_ > 0.07, Figure 2—figure supplement 1A), demonstrating that stimulus drive (as estimated by ERF slope) was approximately equalized over all hues but blue.

The peak time of the N70 component was significantly later for blue stimuli (Mean = 88.6 ms, CI_95%_ = [84.9 ms, 92.1 ms]) compared to all (all p_Holm_ < 0.02) but yellow, green and green-yellow stimuli, for yellow (Mean = 84.4 ms, CI_95%_ = [81.6 ms, 87.6 ms]) compared to red and red-blue stimuli (all p_Holm_ < 0.03), and fastest for red stimuli (Mean = 77.9 ms, CI_95%_ = [74.5 ms, 81.1 ms]) showing a general pattern of slower N70 peaks for stimuli on the S–(L+M) axis, especially for blue (Figure 2—figure supplement 1B).”

We also checked if our main findings (equivalence of drive-controlled red and green stimuli, weaker responses for S+ stimuli) are robust when controlled for differences in ERF parameters and added in the Results:

“To attempt to control for potential remaining differences in input drive that the DKL normalization missed, we regressed out per-participant, per-color, the N70 slope and amplitude from the induced gamma power. Results remained equivalent along the L–M axis: The induced gamma power change residuals were not statistically different between red and green stimuli (Red: 8.22, CI_95%_ = [–0.42, 16.85], Green: 12.09, CI_95%_ = [5.44, 18.75], t(29) = 1.35, p_Holm_ = 1.0, BF_01_ = 3.00).

As we found differences in initial ERF slope especially for blue stimuli, we checked if this was sufficient to explain weaker induced gamma power for blue stimuli. While blue stimuli still showed weaker gamma-power change residuals than yellow stimuli (Blue: –11.23, CI_95%_ = [-16.89, –16.89, –5.57], Yellow: -6.35, CI_95%_ = [-11.20, –11.20, –1.50]), this difference did not reach significance when regressing out changes in N70 slope and amplitude (t(29) = 1.65, p_Holm_ = 0.88). This suggests that lower levels of input drive generated by equicontrast blue versus yellow stimuli might explain the weaker gamma oscillations induced by them.”

We added accordingly in the Discussion:

“The fact that controlling for N70 amplitude and slope strongly diminished the recorded differences in induced gamma power between S+ and S- stimuli supports the idea that the recorded differences in induced gamma power over the S-(L+M) axis might be due to pure S+ stimuli generating weaker input drive to V1 compared to DKL-equicontrast S- stimuli, even when cone contrasts are equalized.”

Additionally, we made the correlation between ERF amplitude and induced gamma power clearer to read by correlating them directly. Accordingly, the relevant paragraph in the results now reads:

“In addition, there were significant correlations between the N70 ERF component and induced gamma power: The extracted N70 amplitude was correlated across colors with the induced gamma power change within participants with on average r = –0.38 (CI95% = [–0.49, –0.28], p_Wilcoxon_ < 4*10^–6^). This correlation was specific to the gamma band and the N70 component: Across colors, there were significant correlation clusters between V1 dipole moment 68-79 ms post-stimulus onset and induced power between 28–54 Hz and 72 Hz (Figure 4C, r_max_ = –0.30, p_Tmax_ < 0.05, corrected for multiple comparisons across time and frequency).”

II. As indicated above, the paper rests on accurate modeling of human LGN recruitment, based in fact on human cone recruitment. However, the exact details of how such matching was obtained were rapidly discussed-this technical detail is much more than just a detail in a study on color matching: I am not against the logic nor do I know of a flaw, but it's the hinge of the paper and is dealt with glancingly.A. Some discussion of model limitations.B. Why it's valid to assume LGN matching has been achieved using data from the periphery: To buy knowledge, nobody has ever recorded single units in human LGN with these color stimuli…in contrast, the ERF is 'in their hands' and could be directly related (or not) to gamma and to the color matching predictions of their model.

We have revised the respective paragraph of the introduction to read:

“Earlier work has established in the non-human primate that LGN responses to color stimuli can be well explained by measuring retinal cone absorption spectra and constructing the following cone-contrast axes: L+M (capturing luminance), L–M (capturing redness vs. greenness), and S-(L+M) (capturing S-cone activation, which correspond to violet vs. yellow hues). These axes span a color space referred to as DKL space (Derrington, Krauskopf, and Lennie, 1984). This insight can be translated to humans (for recent examples, see Olkkonen et al., 2008; Witzel and Gegenfurtner, 2018), if one assumes that human LGN responses have a similar dependence on human cone responses. Recordings of human LGN single units to colored stimuli are not available (to our knowledge). Yet, sensitivity spectra of human retinal cones have been determined by a number of approaches, including ex-vivo retinal unit recordings (Schnapf et al., 1987), and psychophysical color matching (Stockman and Sharpe, 2000). These human cone sensitivity spectra, together with the mentioned assumption, allow to determine a DKL space for human observers. To show color stimuli in coordinates that model LGN activation (and thereby V1 input), monitor light emission spectra for colored stimuli can be measured to define the strength of S–, M–, and L-cone excitation they induce. Then, stimuli and stimulus background can be picked from an equiluminance plane in DKL space.”

Reviewer #2 (Recommendations for the authors):The major strengths of this study are the use of MEG measurements to obtain spatially resolved estimates of gamma rhythms from a large(ish) sample of human participants, during presentation of stimuli that are generally well matched for cone contrast. Responses were obtained using a 10deg diameter uniform field presented in and around the centre of gaze. The authors find that stimuli with equivalent cone contrast in L–M axis generated equivalent gamma – ie. that 'red' (+L–M) stimuli do not generate stronger responses than 'green (–L+M). The MEG measurements are carefully made and participants performed a decrement-detection task away from the centre of gaze (but within the stimulus), allowing measurements of perceptual performance and in addition controlling attention.There are a number of additional observations that make clear that the color and contrast of stimuli are important in understanding gamma. Psychophysical performance was worst for stimuli modulated along the +S–(L+M) direction, and these directions also evoked weakest evoked potentials and induced gamma. There also appear to be additional physiological asymmetries along non-cardinal color directions (e.g. Figure 2C, Figure 3E). The asymmetries between non-cardinal stimuli may parallel those seen in other physiological and perceptual studies and could be drawn out (e.g. Danilova and Mollon, Journal of Vision 2010; Goddard et al., Journal of Vision 2010; Lafer-Sousa et al., JOSA 2012).

We thank the review for the pointers to relevant literature and have added in the Discussion:

“Concerning off-axis colors (red-blue, green-blue, green-yellow and red-yellow), we found stronger gamma power and ERF N70 responses to stimuli along the green-yellow/red-blue axis (which has been called lime-magenta in previous studies) compared to stimuli along the red-yellow/green-blue axis (orange-cyan). In human studies varying color contrast along these axes, lime-magenta has also been found to induce stronger fMRI responses (Goddard et al., 2010; but see Lafer-Sousa et al., 2012), and psychophysical work has proposed a cortical color channel along this axis (Danilova and Mollon, 2010; but see Witzel and Gegenfurtner, 2013).”

Similarly, the asymmetry between +S and -S modulation is striking and need better explanation within the model (that thalamic input strength predicts gamma strength) given that +S inputs to cortex appear to be, if anything, stronger than -S inputs (e.g. DeValois et al. PNAS 2000).

We followed the reviewer’s suggestion and modified the Discussion to read:

“Contrary to the unified pathway for L–M activation, stimuli high and low on the S–(L+M) axis (S+ and S–) each target different cell populations in the LGN, and different cortical layers within V1 (Chatterjee and Callaway, 2003; De Valois et al., 2000), whereby the S+ pathway shows higher LGN neuron and V1 afferent input numbers (Chatterjee and Callaway, 2003). Other metrics of V1 activation, such as ERPs/ERFs, reveal that these more numerous S+ inputs result in a weaker evoked potential that also shows a longer latency (our data; Nunez et al., 2021). The origin of this dissociation might lie in different input timing or less cortical amplification, but remains unclear so far. Interestingly, our results suggest that cortical gamma is more closely related to the processes reflected in the ERP/ERF: Stimuli inducing stronger ERF induced stronger gamma; and controlling for ERF-based measures of input drives abolished differences between S+ and S– stimuli in our data.”

Given that this asymmetry presents a potential exception to the direct association between LGN drive and V1 gamma power, we have toned down claims of a direct input drive to gamma power relationship in the Title and text and have refocused instead on L–M contrast.

My only real concern is that the authors use a precomputed DKL color space for all observers. The problem with this approach is that the isoluminant plane of DKL color space is predicated on a particular balance of L- and M-cones to Vlambda, and individuals can show substantial variability of the angle of the isoluminant plane in DKL space (e.g. He, Cruz and Eskew, Journal of Vision 2020). There is a non-negligible chance that all the responses to colored stimuli may therefore be predicted by projection of the stimuli onto each individual's idiosyncratic Vlambda (that is, the residual luminance contrast in the stimulus). While this would be exhaustive to assess in the MEG measurements, it may be possible to assess perceptually as in the He paper above or by similar methods. Regardless, the authors should consider the implications – this is important because, for example, it may suggest that important of signals from magnocellular pathway, which are thought to be important for Vlambda.

We followed the suggestion of the reviewer, performed additional analyses and report the new results in the following Results text:

“When perceptual (instead of neuronal) definitions of equiluminance are used, there is substantial between-subject variability in the ratio of relative L- and M-cone contributions to perceived luminance, with a mean ratio of L/M luminance contributions of 1.5-2.3 (He et al., 2020). Our perceptual results are consistent with that: We had determined the color-contrast change-detection threshold per color; We used the inverse of this threshold as a metric of color change-detection performance; The ratio of this performance metric between red and green (L divided by M) had an average value of 1.48, with substantial variability over subjects (CI95% = [1.33, 1.66]).

If such variability also affected the neuronal ERF and gamma power measures reported here, L/M-ratios in color-contrast change-detection thresholds should be correlated across subjects with L/M-ratios in ERF amplitude and induced gamma power. This was not the case: Change-detection threshold red/green ratios were neither correlated with ERF N70 amplitude red/green ratios (ρ = 0.09, p = 0.65), nor with induced gamma power red/green ratios (ρ = –0.17, p = 0.38).”

Please show sign/scale on axes in Figure 2C/D and similar for legibility.

Axis scale labels have been added to Figure 2C/D and all other rose plots. As an example, see Figure 2.

Please report the cone contrast values of the stimuli that were presented.

Cone contrast coordinates have been added to Figure 1—figure supplement 1.

Reviewer #3 (Recommendations for the authors):This is an interesting article studying human color perception using MEG. The specific aim was to study differences in color perception related to different S-, M-, and L-cone excitation levels and especially whether red color is perceived differentially to other colors. To my knowledge, this is the first study of its kind and as such very interesting. The methods are excellent, and manuscript is well written as expected this manuscript coming from this lab. However, illustrations of the results is not optimal and could be enhanced.The results presented in the manuscript are very interesting, but not presented comprehensively to evaluate the validity of the results. The main results of the manuscript are that the gamma-band responses to stimuli with absolute L-M contrast i.e. green and red stimuli do not differ, but they differ for stimuli on the S-(L+M) (blue vs red-green) axis and gamma-band responses for blue stimuli are smaller. These data are presented in figure 3, but in its current form, these results are not well conveyed by the figure. The main results are illustrated in figures 3BC, which show the average waveforms for grating and for different color stimuli. While there are confidence limits for the gamma-band responses for the grating stimuli, there are no confidence limits for the responses to different color stimuli. Therefore, the main results of the similarities / differences between the responses to different colors can't be evaluated based on the figure and hence confidence limits should be added to these data.

Figure 3E reports the gamma-power change values after alignment to the individual peak gamma frequencies, i.e. the values used for statistics, and does report confidence intervals. Yet, we see the point of the reviewer that confidence intervals are also helpful in the non-aligned/complete spectra. We found that inclusion of confidence intervals into Figure 3B,C, with the many overlapping spectra, renders those panels un-readable. Therefore, we included the new panel Figure 3—figure supplement 2A, showing each color’s spectrum separately:

It is also not clear from the figure legend, from which time-window data is averaged for the waveforms.

We have added in the legend:

“All panels show power change 0.3 s to 1.3 s after stimulus onset, relative to baseline.”

The time-resolved profile of gamma-power changes are illustrated in Figure 3D. This figure would a perfect place to illustrate the main results. However, of all color stimuli, these TFRs are shown only for the green stimuli, not for the red-green differences nor for blue stimuli for which responses were smaller. Why these TFRs are not showed for all color stimuli and for their differences?

We agree with the reviewer that TFR plots can be very informative. We followed their request and included TFRs for each color as Figure 3—figure supplement 3.

Regarding the suggestion to also include TFRs for the differences between colors, we note that this would amount to 28 TFRs, one each for all color combinations. Furthermore, while gamma peaks were often clear, their peak frequencies varied substantially across subjects and colors. Therefore, we based our statistical analysis on the power at the peak frequencies, corresponding to peak-aligned spectra (Figure 3c). A comparison of Figure 3C with Figure 3B shows that the shape of non-aligned average spectra is strongly affected by inter-subject peak-frequency variability and thereby hard to interpret. Therefore, we refrained from showing TFR for differences between colors, which would also lack the required peak alignment.

All main results are now only shown for data averaged over V1. Taken that V4 that is traditionally associated with color processing, the same results could also be shown for this region.

We report, in the Results text and in Figure 3H, the overall color-induced gamma-power change values for all early visual areas, including V4. This revealed that overall color-induced gamma is much weaker in V4 than V1. This spatial pattern might be consistent with one prominent source in V1 whose source-projected power is partially smeared over early visual areas. This precludes a meaningful analysis of V4-projected data and a fair comparison between V4 and V1 in terms of color differences. We agree with the reviewer that the prominent role of V4 in the color literature requires a discussion of this point, and we have therefore added the following to our Discussion section:

“In our data, the color-induced gamma oscillation clearly localized to early visual cortex, was narrow-banded and showed (for 90% of subjects) a second spectral peak around 80-100 Hz, which did not seem to be a harmonic of the main induced peak and which was rare for grating stimuli. As the induced gamma-power change source-localized as a continuous source over the early visual cortex with a peak in V1, our data cannot differentiate between a strong gamma source in V1 that shows source smearing over neighboring areas and, alternatively, several gamma sources in V1-V4 that show monotonous decreases in power with distance from V1.”

However, our finding of stronger color gamma responses in V1 versus V4 were mirrored in a macaque study, and we have added in the Discussion to point out this fact:

“In an initial report on recordings in anesthetized macaque V1 and V4, Rols et al., (2001) first described stronger gamma power for a red stimulus compared to a green and a yellow one (*heteroluminant*, all presented on a gray background). Similar to our study, color-induced gamma oscillations were significantly stronger in V1 than in V4 in their recordings.”

The authors are in results referring to a second higher gamma peak that was detected in 90% of participants for color but not for grating stimuli. How was the within-subject statistics performed to evaluate the presence of this peak?

We have clarified in the Results:

“To test for the existence of gamma peaks, we fit the per-participant, per-stimulus change spectra with a) the sum of two gaussians and a linear slope, b) the sum of one Gaussian and a linear slope and c) only a linear slope (without any peaks); subsequently, we chose the best-fitting model using adjusted R^2^-values.”

The authors discuss the localization accuracy of source-localized MEG. In addition to rough approximations, the authors could also evaluate the specific source localization accuracy of their method in their specific subject population by using simulated data together with realistic forward and inverse modelling. Taken that V1 as a medial structure might not be well source-modelled, this could give important insight on the factors influencing the results.

Three arguments give us confidence that our source recordings are from early visual cortex, with a probable focus in V1. Previous work from our lab has shown high levels of similarity in findings from macaque V1-V4 recordings and MEG recordings source-localized to the same areas (Bastos et al., 2015; Michalareas et al., 2016; Peter et al., 2021; Stauch et al., 2021). Additionally, both our ERF and gamma power responses to colored stimuli agree well with previous reports in intracranial macaque recordings (Peter et al., 2019; Rols et al., 2001). Thirdly, V1 has been found to be well picked up as a gamma source and as a source of visually induced signals in other MEG (Duecker et al., 2021; Hall et al., 2005; Hoogenboom et al., 2006; Nasiotis et al., 2017) and EEG studies (Butler et al., 2019).

As source localization was computed (and dipole orientation was fixed) over all stimulus condition, it could not explain our results concerning differences between different stimuli.

The correlation of the gamma-power but not ERF with behavioral performance is interesting. The authors might want to elaborate and refer to Julku et al., 2021 Sci. Rep. which found a correlation between oscillations and EDRTs values but not with EFRs and EDTRs values.

There seems to be a misunderstanding: In our hands, behavioral performance (as measured using the staircase outcome) correlates with both gamma power and ERF amplitude, as described in the Results:

“Higher change detection performance (defined as lower final staircased target color contrast) was correlated with stronger average V1 dipole moment (i.e. the negative ERF amplitude during the N70 component) across colors, in the time period from 67 to 93 ms post stimulus onset (Figure 4A, r_max_ = 0.43, all p_Tmax_ < 0.03, corrected for multiple comparisons across time). Higher change detection performance was also correlated with stronger induced V1 power across colors, for several gamma frequency ranges (Figure 4B; at 34–37 Hz: r_max_ = –0.27, all p_Tmax_ <= 0.03; at 39–40 Hz: r_max_ = –0.23, all p_Tmax_ < 0.04; at 44–53 Hz: r_max_ = –0.31, all p_Tmax_ <= 0.03, corrected for multiple comparisons across frequency).”